# Antibodies targeting ADAM17 reverse neurite outgrowth inhibition by myelin-associated inhibitors

Nayanendu Saha[1], Eric Chan[2], Rachelle P Mendoza[3], Yevgeniy Romin[2], Murray J Tipping[2], Dimitar B Nikolov[1]

Upon spinal cord injury, axons attempting to regenerate need to overcome the repulsive actions of myelin-associated inhibitors, including the myelin-associated glycoprotein, Nogo-A, and the oligodendrocyte myelin glycoprotein. These inhibitors bind and signal through a neuronal receptor/co-receptor/transducer complex composed of NgR1, Lingo-1, and p75. Consequently, p75 is cleaved by alpha secretase followed by gamma-secretase, triggering downstream signaling that inhibits axonal regrowth. ADAM10 and ADAM17 are both known to function as alpha secretases in neurons. Here we show that ADAM17, and not ADAM10, is the alpha secretase that recognizes and cleaves p75, when it is a part of a 5-component neuron-myelin signaling complex comprising NgR1, Lingo-1, p75, GT1b, and a myelin inhibitor. Importantly, we demonstrate the ability of inhibitory anti-ADAM17 mAbs to abrogate the cleavage of p75 in a neuroblastoma-glioma cell line and reverse the neurite outgrowth inhibition by myelin-associated inhibitors.

## Introduction

The central nervous system axons, attempting to regenerate after mechanical damage or injury, face major hurdles. The glial scar formed at the site of the lesion physically hinders regeneration (1). Importantly, the axons have to overcome the repulsive action of myelin-associated inhibitors, such as the myelin-associated glycoprotein (MAG), Nogo-A, and the oligodendrocyte myelin glycoprotein (2). These inhibitors interact with and signal through a neuronal receptor/co-receptor/transducer complex comprising NgR1, Lingo-1, and p75, respectively. Previously, we showed that gangliosides, and specifically GT1b, mediate the binding of NgR1 to Lingo-1 and their complex then binds to the p75 and the myelin inhibitor Nogo-A (3, 4).

Upon binding of a myelin-associated ligand to the neuronal receptor/co-receptor complex, p75 undergoes regulated intramembrane proteolysis (RIP) by alpha secretase followed by gamma-secretase (5). This leads to the activation of Rho-associated coiled-coil–containing kinase (RhoA/ROCK). As a sequel, downstream effector molecules associated with cytoskeletal reorganization are activated and bring forth growth cone collapse and neurite outgrowth inhibition (6). It has been shown that several ADAM family metalloproteases are well expressed in the central nervous system, and that ADAM10 and ADAM17 can cleave p75, thus potentially fulfilling the role of alpha secretase (7). However, the molecular details of the interactions of the neuronal receptor, co-receptor (NgR1, Lingo-1) complex with the myelin-based ligands, and the resulting initiation of p75-mediated signaling are still not known. Likewise, the interaction of p75 with ADAM proteases has not been studied.

ADAM metalloproteases contain several conserved protein domains: an N-terminal signal sequence is followed by a prodomain, metalloprotease (MP) domain, disintegrin (D) domain, cysteine-rich (C) domain (an EGF-like domain is also suggested for some family members), transmembrane domain, and cytoplasmic domain (8). ADAM proteases release extracellular domains of membrane-tethered proteins in a regulated, substrate-specific manner, yet without a strong preference for a cleavage sequence signature. The ADAMs can mediate both constitutive and regulated shedding of cell-surface ligands (9, 10, 11). At present, little is known about how the ADAMs recognize their substrates. Our previous studies have highlighted the importance of the ADAM D and C domains in substrate recognition. Specifically, we have shown that the D and C (D+C) domain region mediates interactions of ADAM proteases (in particular ADAM17 and ADAM10) with substrates, such as Eph-receptor/ephrin complexes and Notch receptors (12, 13, 14, 15, 16, 17). Furthermore, we have raised panels of ADAM17- and ADAM10-specific inhibitory mAbs that disrupt the binding of these ADAM proteases to their substrates, thus inhibiting proteolysis. These anti-ADAM mAbs also deter the proliferation of a range of cancer cells that depend on ADAM-mediated signaling both in vitro and in vivo (in mouse xenograft models) (17, 18).

## Results and Discussion

### The neuronal receptor NgR1 binds the co-receptor Lingo-1 in the presence of GT1b and then forms a stable 5-mer complex with the myelin inhibitor MAG and the signal transducer p75

The molecular signaling pathways in the nervous system, which result in the inhibition of neuronal regrowth upon injury, involve

[1]Structural Biology Program, Memorial Sloan Kettering Cancer Center, New York, NY, USA   [2]Molecular Cytology Core Facility, Memorial Sloan Kettering Cancer Center, New York, NY, USA   [3]Department of Pathology and Cell Biology, Columbia University Irving Medical Center, New York, NY, USA

Correspondence: sahan@mskcc.org; nikolovd@mskcc.org

ligand-induced RIP of the signal transducer p75 (1, 2). The signaling events after the release of the cleaved intracellular fragment of p75 have been well characterized (5, 6), whereas the steps before p75 cleavage, including the assembly of the NgR1, Lingo-1, p75 complex at the neuronal surface, remain unknown at large. To that end, we previously reported the important role of the trisialoganglioside GT1b, an integral component of the neuronal membrane, in the formation of the NgR1, Lingo-1 complex at the neuronal and axonal cell surface. We further showed that, once formed, this receptor, co-receptor complex, can bind to p75 and Nogo-A (3, 4).

Here, using purified proteins, we substantiated our findings using a different myelin-associated inhibitor, MAG. We first performed a biochemical pull-down experiment using protein A Sepharose. We immobilized a Lingo-Fc construct (residues 35–516) on protein A beads and pulled-down a mixture of NgR1 and p75 (residues 31–210; this construct behaves the same in the pull-down assays as the 31–250 construct but is easier to separate on the SDS gels from the NgR1 construct). We found that their binding was contingent on the presence of GT1b-Na, reaffirming that these interactions are ganglioside mediated. For our studies, we used an NgR1 construct (residues 30–430) containing the LRR repeats and the stalk region (connecting the LRRs to the membrane) that we previously showed was required for GT1b binding (3) (Fig 1B).

To further observe the formation of the signaling complex in solution, the neuronal receptor, co-receptor, p75 complex components, NgR1, Lingo-1, and the p75 ectodomain (residues 31–250) were also mixed at a molar ratio of 1:1:1 in the presence or absence of GT1b-Na salt and were subjected to size-exclusion chromatography on a SD-200 column. The peak containing the complex eluted at 440 kD (with a similar Kav and elution profile as the control marker Ferritin) in the presence of GT1b-Na. Specifically, the elution volume was 10.5 ml (fraction 11) in the presence of GT1b-Na, compared with 12.5 ml in the absence of the ganglioside. SDS–PAGE analysis confirmed that, in the presence of GT1b, the complex included NgR1, Lingo-1, and p75 (Fig 1D). Individually, Lingo-1 migrates as a tetramer, Nogo receptor as a monomer, and p75 as an apparent dimer (Fig 1C). These results agree with the previously reported oligomerization states observed in the crystal structures of the Lingo-1 and NgR1 ectodomains (19, 20, 21, 22). The structure of unbound p75 has not yet been reported, although the complex of neurotrophin-3 (NT-3) and p75 revealed a 2:2 heterodimer (23, 24). These data reinstate that GT1b-Na facilitates the assembly of the neuronal receptor, co-receptor, p75 complex.

We also documented, using our pull-down assay, that the NgR1, Lingo-1, p75 complex, in the presence of GT1b-Na, binds the myelin-associated inhibitor MAG (residues 22–504) (Fig 1B). The resulting biochemically stable complex, which comprises NgR1, Lingo-1, p75, GT1b-Na, and MAG, is referred here as the "neuron-myelin 5-mer signaling complex."

## ADAM17, and not ADAM10, interacts with p75 and with the neuron-myelin 5-mer signaling complex

The identity of the ADAM protease that functions as the alpha secretase and cleaves p75 upon axonal/neuronal damage is still unclear. Dissecting the interactions of the alpha secretase with the neuron-myelin signaling complex is pivotal for understanding the

molecular events underlying spinal cord injury and paralysis and for the development of drugs promoting neuronal regeneration. Previous studies revealed that ADAM17 affects the stimulated shedding of p75 in Chinese hamster ovary cells (CHO). Indeed, in mutant CHO cells that lack ADAM17 but express ADAM9, ADAM10, ADAM12, and ADAM15, the shedding of p75 was completely abrogated (25). However, ADAM10 has also been reported to be involved in RIP of p75, specifically during processes that aid cancer cells in acquiring resistance to programmed cell death (26).

Using pull-down experiments, ELISA-based binding assays, and Bio-Layer Interferometry (BLI), we now evaluated the in vitro interactions of ADAM17 and ADAM10, with the 5-mer (NgR1, Lingo-1, p75, MAG, GT1b) neuron-myelin signaling complex and with isolated p75. First, we performed ELISA-based assays to gauge the binding of the purified 5-mer complex (purification is described in detail in Materials and Methods section, Fig 2A), or of the isolated p75, to the immobilized ADAM10 ectodomain (ECD), residues 214–646, or the ADAM17 ectodomain (ECD), residues 215–650, or the ADAM disintegrin + cysteine-rich domain region, ADAM10 D+C (residues 455–646) or ADAM17 D+C (residues 484–642). For the ADAM ECD constructs, we used the active-site mutants ADAM17 (E406A) and ADAM10 (E384A) to prevent any unwarranted proteolysis (8, 9). The results showed that ADAM10 does not bind to the 5-mer complex or to isolated p75 (Fig 2B and C). However, the 5-mer complex did bind to both ADAM17 ECD and ADAM17 D+C, whereas p75 bound only to the isolated ADAM17 D+C region. Thus, the binding of ADAM17 to the 5-mer is most likely mediated by the D+C domains, which are known to be involved in determining substrate specificity (12, 13, 14, 15, 16). These results were further corroborated by BLI using the instrument Blitz (Forte Bio). The ADAM17 ECD construct bound to the 5-mer complex with an apparent KD of ~500 nM, whereas the ADAM17 D+C construct bound to the 5-mer complex and the isolated p75 with a KD of ~350 nM; the ADAM17 ECD did not bind to isolated p75 (Fig S1A–D). The results thus highlighted that the ADAM17 complete ectodomain interacts with p75 only in the context of the assembled neuron-myelin signaling complex. The interactions of ADAM17 with the GT1b-mediated neuronal receptor/co-receptor/p75 complex, and with the 5-mer neuron-myelin signaling complex, were further verified by biochemical pull-down experiments using the ADAM17 ECD or D+C protein constructs. The data (Fig 3A and B) confirm that both ADAM17 ECD and the isolated D+C domains interact with the neuronal receptor/co-receptor/p75 complex and the neuron-myelin 5-mer complex. The pull-down experiments also corroborated that ADAM17 ECD does not bind to isolated p75. The ADAM17 D+C domain construct, however, as observed in ELISA-based and BLI experiments, interacts with p75 (Fig 3C and D).

The results outlined above document that ADAM17, and not ADAM10, is the alpha secretase that cleaves p75 in the context of spinal cord injury, as only ADAM17 interacts with the neuron-myelin signaling complex and with p75. This interaction is mediated by the ADAM17 D+C domain region. Interestingly, the full ADAM17 ECD does not bind p75 if it is not a part of the signaling complex, suggesting that this complex triggers a conformation change in ADAM17 where the D+C region becomes accessible for interaction with p75. Indeed, it has previously been postulated that ADAM proteinases sample two distinct conformations: open/activated, where the D+C region

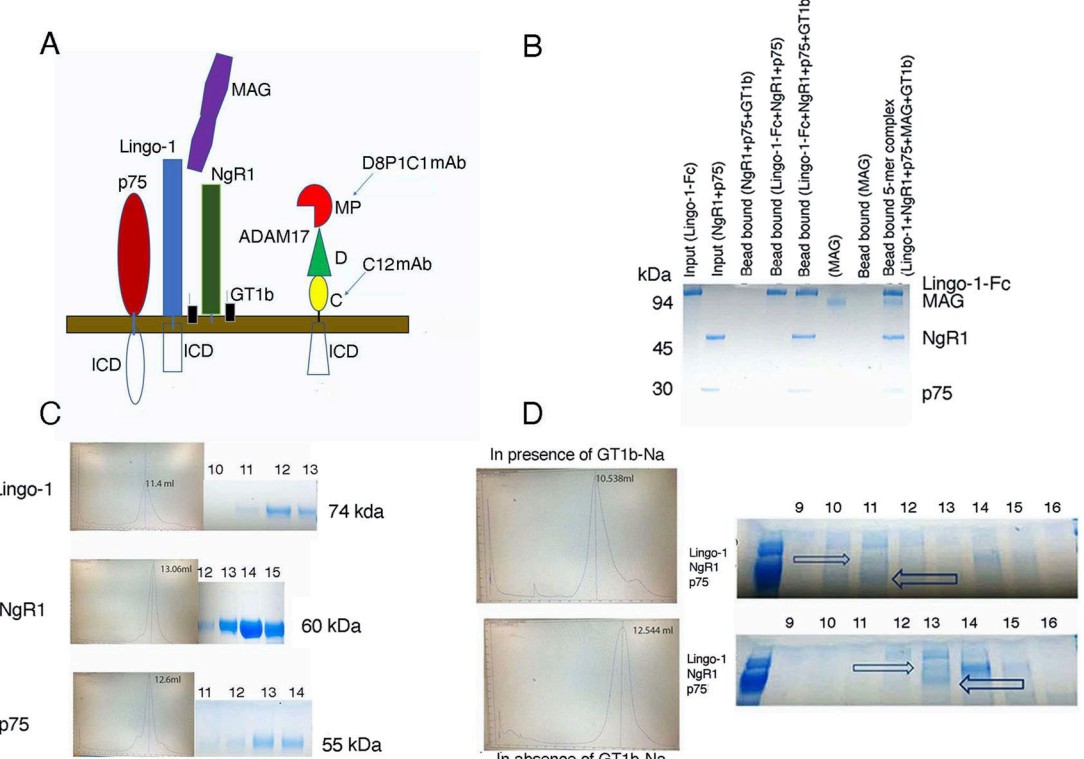

**Figure 1. GT1b-mediated formation of the neuronal-receptor/co-receptor/transducer (NgR1, Lingo-1, p75) and the neuron-myelin (NgR1, Lingo-1, p75, MAG) signaling complexes.**
**(A)** Schematic representation of the neuronal receptor and co-receptor, the myelin inhibitor, and ADAM17 mediating the inhibitory signaling upon spinal cord injury. The colored parts represent the soluble, extracellular constructs that were used in this study and arrowheads denote the binding epitopes of the anti-ADAM17 mAbs. ICD denotes the intracellular domains. **(B)** Biochemical pull-down assay using protein A-Sepharose beads. Inputs: Lingo-Fc (10 µg); NgR1: (6 µg); p75: (4 µg); MAG: (8 µg). Where indicated, trisialoganglioside was added as a GT1b-Na salt. NgR1 and p75 were pre-incubated with GT1b-Na. Lingo-1-Fc was used as a "bait," whereas the untagged NgR1, p75, and MAG proteins were used as "prey." The protein A-Sepharose–bound fractions were analyzed by SDS–PAGE. **(C)** Gel Filtration (on a SD-200 column) profiles of Lingo-1, NgR1, and p75. The numbers above the gel lanes indicate the fraction numbers (each fraction is 1 ml). **(D)** Gel Filtration (on a SD-200 column) profiles of the NgR1, Lingo-1, p75 complex in the presence and absence of GT1b-Na. The elution volumes are listed above the peaks. The forward and reverse arrowheads denote the migration of NgR1 and p75, respectively, in the presence and absence of GT1b-Na salt. In the presence of GT1b-Na, all three proteins form a complex that elutes at 10.5 ml. The fractions from SD-200 were run on the SDS–PAGE. The numbers represent the different fractions from SD-200.

is exposed for binding to substrates and regulatory proteins, and closed/autoinhibited, where the D+C region interacts with and inhibits the ADAM catalytic (MP) domain (16). In addition, recent structural studies with the mature form of the ADAM17 ectodomain bound to iRhom2 showed a high degree of conformational flexibility in ADAM17 (27). We, therefore, hypothesize that the ADAM17 interactions with the neuron-myelin signaling complex abet the mature alpha secretase to attain and maintain an activated/open conformation, leading to the RIP of p75. Such an activation mechanism would preclude the cleavage of p75 molecules that are not associated with signaling complexes, thus preventing aberrant downstream signaling. The activation events might be facilitated by iRhoms (27), e.g., iRhom1, known to be expressed in the mouse brain and to regulate ectodomain shedding by ADAM17. The reported here isolation of a biochemically stable 5-mer (NgR1, LINGO-1, GT1b, p75, MAG) signaling complex (Fig 2A) could also facilitate future high-resolution structural studies of the initiation of the neuron-myelin inhibitory signaling during spinal cord injury and regeneration.

## Anti–ADAM17 mAbs inhibit the interactions between ADAM17 and the neuron-myelin signaling complex

We evaluated several mAbs targeting different domains of ADAM17 for their ability to disrupt the ADAM17 interactions with the neuron-myelin 5-mer complex and to impede the downstream signaling cascade that inhibits neuronal regeneration. These included two affinity-matured mAbs, D8P1C1, and D5P2A11, described by us before (18), which bind to the protease domain. In addition, we used a more recently generated mAb, C12, which binds to the cysteine-rich domain of ADAM17 (28). First, using an ELISA-based assay, we examined the effects of the mAbs on the binding of the neuron-myelin 5-mer signaling complex to immobilized ADAM17 ECD (Fig 4A) and D+C constructs (Fig 4B). These studies showed that C12 partially inhibits the binding of the ADAM17 ECD and D+C to the 5-mer complex, whereas D8P1C1 and D5P2A11 have no effect, suggesting that the protease domain is not involved in significant interactions with this particular ADAM17 substrate, the 5-mer signaling complex.

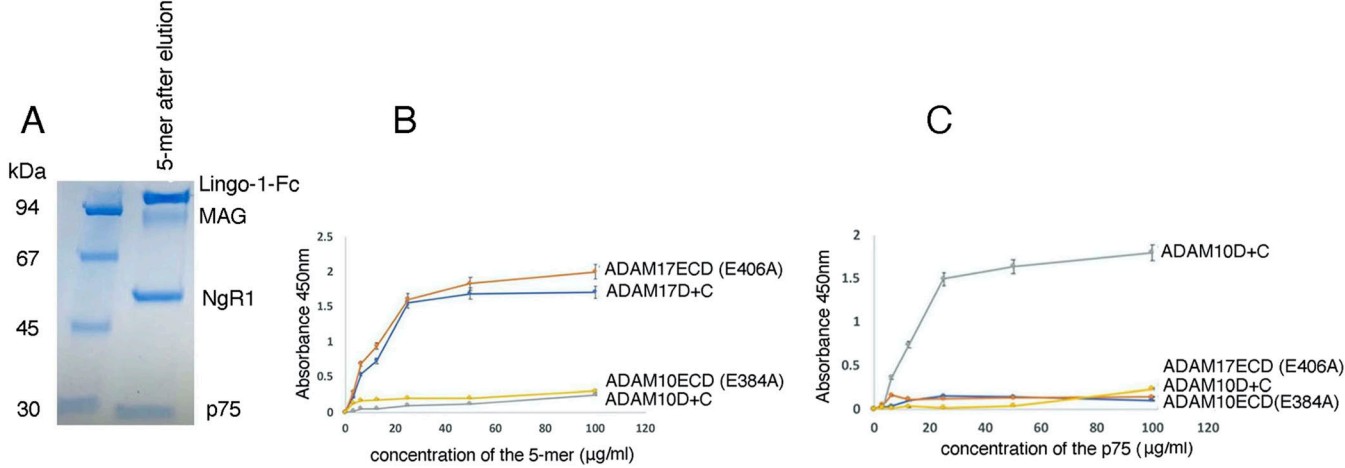

**Figure 2. Interactions of ADAM17 and ADAM10 with the neuron-myelin signaling complex.**
**(A)** Purification of the 5-mer neuron-myelin signaling complex for ELISA-based studies. The 5-mer complex was formed in the presence of GT1b-Na as described in Fig 1. Briefly, bead-bound Lingo-1-Fc was used to pull-down NgR1, p75, and MAG in the presence of GT1b-Na. The beads were gently washed, and the bound proteins were eluted with low pH (100 mM glycine-HCl, pH 3.5). After elution, the pH was raised to 7 and the complex was analyzed on SDS–PAGE. **(B, C)** ELISA-based assay to gauge the binding of the 5-mer neuron-myelin complex (B) or p75 (C) to immobilized ADAM17 or ADAM10 (ECD or D+C protein constructs). **(A)** Briefly, wells were coated with ADAM17 ECD (E406A) or ADAM17 D+C or ADAM10 ECD (E384) or ADAM10 D+C, and varying concentrations of the 5-mer complex (isolated as described in panel (A)) or p75 were added to the wells and incubated for 1 h at RT. The bound p75 was detected by mouse mAb specific to the ectodomain of human p75. To detect the bound 5-mer complex, mouse mAb to the MAG (human) ectodomain was used. Data (n = 3) was recorded at 450 nm.

### Anti–ADAM17 mAbs abrogate p75 shedding in a neuronal model system

Using differentiated NG108-15 neuroblastoma-glioma cells, we evaluated the effect of the anti-ADAM17 mAbs on the release of cleaved p75 ectodomain in the conditioned media. It has been shown that upon RIP, an ~55 kD fragment from the cell-tethered p75 is released in the extracellular milieu (25). We first confirmed the expression of NgR1, Lingo-1, and p75 on the cell surface of the NG108-15 cells. Significant expression of NgR1, Lingo-1, and p75 was observed only after differentiating the cells with 1 mM dibutyl cAMP (29) (Fig 5A). Differentiation of NG108-15 has been previously shown to be necessary for the expression of neuronal proteins (31). Before evaluating the potency of the anti-ADAM17 mAbs to deter myelin-induced shedding of p75, we determined the optimal conditions for the assay. Specifically, we observed that maximum shedding of the p75 ectodomain occurs in the presence of PMA (phorbol myristate acetate) (25 ng/ml, 30′ of incubation) followed by the addition of the myelin-associated inhibitor MAG (20 μg/ml, additional 30′ of incubation) (Fig 5B). PMA is a non-physiological PKC activator, which has been shown to augment ADAM17 cell-surface activity (25). The shedding of the p75 ectodomain is inhibited by the anti-ADAM17 small-molecule inhibitor TAPI-1 (30) both in the absence (Fig 5C) and presence (see Fig 6B) of PMA.

Next, we evaluated how administration of the anti-ADAM mAbs influences the MAG-induced release of the 55 kD p75 fragment in the media. We also included small-molecule inhibitors that block the proteolytic activity of either ADAM10, ADAM17 or matrix metalloproteases. A control experiment was performed in the absence of any inhibitor. The intensity of the band (55 kD fragment in the supernatant) was quantitated using the Image J software. GAPDH was used as a loading control. For our experiment, we used

the three anti-ADAM17 mAbs, D8P1C1 (18), D5P2A11 (18) and C12 (28). The results highlighted that the D8P1C1 and D5P2A11 mAbs, which target the protease active site, are the most potent inhibitors of p75 shedding (Fig 6A, C, and D). The C12 mAb shows significant inhibition 95% at the highest concentration tested (20 μg/ml), and though, unlike D8P1C1 and D5P2A11, its inhibitory effect wears off at lower concentrations (Fig 6B–D). TAPI-1, an anti-ADAM17 small-molecule inhibitor (30), also showed a moderate effect (70% inhibition) at 1 μM (Fig 6B and D). On the contrary, neither the anti-ADAM10 inhibitory mAb 1H5 (17) that binds to the substrate-binding domain, nor the anti-ADAM10 small-molecule inhibitor GI 254023X (32), had any effect, suggesting that ADAM10 is not involved in p75 cleavage in NG108-15 cells. Batimastat (33), a broad-spectrum inhibitor of matrix metalloproteinase (MMP), was also ineffective in deterring cleavage of p75 (Fig 6B and D). These results establish that, in the neuroblastoma cell line most-commonly used to assay p75 shedding in vitro, ADAM17 is the primary alpha secretase that initiates RIP of p75, en route to initiating inhibitory downstream signaling.

### Anti–ADAM17 mAbs suppress MAG-induced neurite outgrowth inhibition

We evaluated the ability of the anti-ADAM17 inhibitors to reverse the inhibition of neurite outgrowth in differentiated NG108-15 cells (treated with PMA) upon addition of MAG. The assay we used here has been described previously (34, 35). A control with no added MAG was used as a reference. The data show that the anti-ADAM17 mAbs that bind to the protease domain and block the cleavage of p75 were the most effective in blocking the MAG-induced neurite outgrowth inhibition (Fig 7A and B, Tables S1 and S2). ADAM10 inhibition by the small-molecule inhibitor GI254023X, or the fully

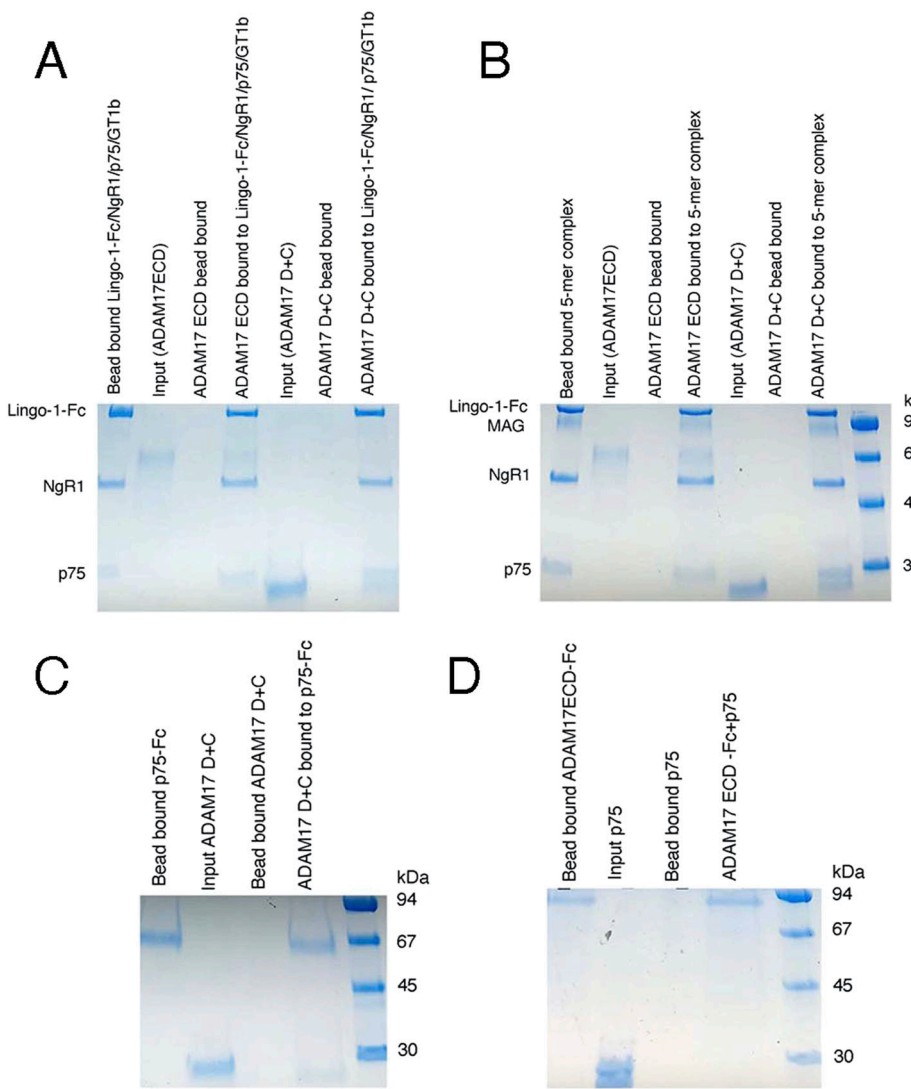

**Figure 3. Biochemical pull-down experiments.**
**(A)** Binding of ADAM17 ECD (E406A) or ADAM17 D+C constructs to the neuronal receptor/co-receptor/p75 complex in the presence of GT1b-Na. Protein A-Sepharose pull-down assays were used to detect the binding of the ADAM17 ECD (active-site mutant E406A) (4 µg) or ADAM17 D+C (8 µg). The neuronal complex comprising Lingo-1-Fc, NgR1, and p75 was pre-formed in the presence of GT1b and used as a "bait." The "prey" ADAM17 protein constructs were added, incubated, washed twice, and analyzed by SDS–PAGE. **(B)** Binding of ADAM17 ECD and ADAM17 D+C to the 5-mer neuron-myelin complex. Here, the pre-formed 5-mer complex was used as a "bait" to pull-down ADAM17 ECD or ADAM17 D+C. **(C)** Binding of ADAM17 D+C to isolated p75. The binding of ADAM17 D+C to Fc-tagged p75 was gauged using protein A-Sepharose beads with p75-Fc (4 µg) immobilized on the beads. The candidate ADAM17 D+C (8 µg) protein was added, washed, and bound proteins were detected by SDS–PAGE. **(D)** Binding of p75 to ADAM17 ECD. Because ADAM17 ECD and p75-Fc show identical mobility on SDS–PAGE (reducing conditions), we performed the experiment using untagged p75 (10 µg) as "prey" and ADAM17 ECD (E406A)-Fc (3 µg) immobilized on protein A-Sepharose beads.

human mAb 1H5 (17) that binds to the D+C domains and prevents substrate access, had no effect, just as MMP inhibition by Batimastat.

In summary, the studies presented above use an array of small-molecule inhibitors and mAbs, which are specific to ADAM17, ADAM10 or MMPs, to clearly identify ADAM17 as the neuronal alpha secretase responsible for p75 cleavage and myelin-induced inhibitory signaling. The results agree with previous observations that blocking p75 cleavage using small-molecule gamma-secretase inhibitors, deters the inhibitory signaling and promotes neurite growth (34). In addition, we show that ADAM17 can only interact with and cleave p75 when it is a part of the neuron-myelin signaling complex, thus preventing aberrant downstream signaling initiation by RIP of p75 molecules outside of signaling complexes. Our model is that the ADAM17 interaction with the neuron-myelin signaling complex promotes its transition to an activated/open conformation, allowing for RIP of p75. This might represent the first reported case of substrate-induced conformational rearrangement/activation of an ADAM proteinase. Interestingly, whereas the alpha secretase conformational

rearrangement is MAG independent (Fig 3), MAG is needed for affecting the cleavage (Fig 5). Elucidating the exact mechanism of how MAG facilitates the cleavage of p75 within the neuron-myelin signaling complex requires further structural studies.

We also characterize candidate therapeutic anti-ADAM17 antibodies that selectively disrupt the interactions of the alpha secretase with the neuron-myelin signaling complex, abrogate shedding of p75 in the neuroblastoma-glioma cell line NG108-15 and, most importantly, reverse the MAG-induced neurite outgrowth inhibition. The antibodies that directly bind to the catalytic domain of ADAM17 and block its proteolytic activity were the most effective among the inhibitors evaluated in this study.

Previous efforts to target the Nogo receptor pathway to promote neuronal regeneration after spinal cord injury have yielded indecisive results. For example, studies that targeted Nogo-A with anti-Nogo-A mAbs, including AT1355, 01424423, and 1435993 (Novartis LLC), remained inconclusive. Likewise, GSK LLC conducted clinical trials with the anti-MAG mAb GSK249320 in 2011, but the final results are yet to be announced. To date, no anti-NgR1 mAbs have demonstrated

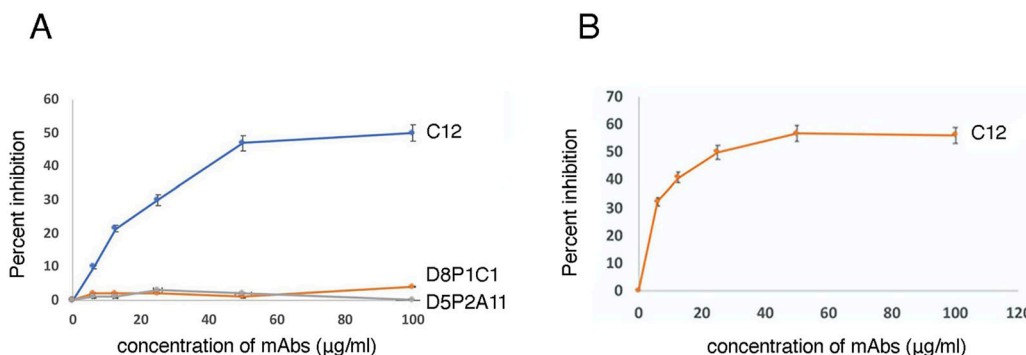

**Figure 4. ELISA-based assay to gauge the effect of three anti-ADAM17 mAbs, D8P1C1, D5P2A11 and C12, on the binding of the 5-mer neuron–myelin complex to ADAM17.**
**(A)** Effect of the mAbs on the binding of the neuron–myelin complex to immobilized ADAM17 ECD (E406A). As before (Fig 2B and C), wells were coated with ADAM17 ECD (E406A), overnight at 4°C. As indicated in the figure, varying concentrations of the three mAbs (D8P1C1, D5P2A11, and C12) were added to the wells and the 5-mer complex bound to ADAM17 was detected by mouse mAb to MAG (human) ectodomain. Data (n = 3) were recorded at 450 nm. **(B)** Effect of the C12 mAb on the binding of the neuron–myelin complex to immobilized ADAM17 D+C. The wells were coated with ADAM17 D+C and the assay was executed as described in panel (A).

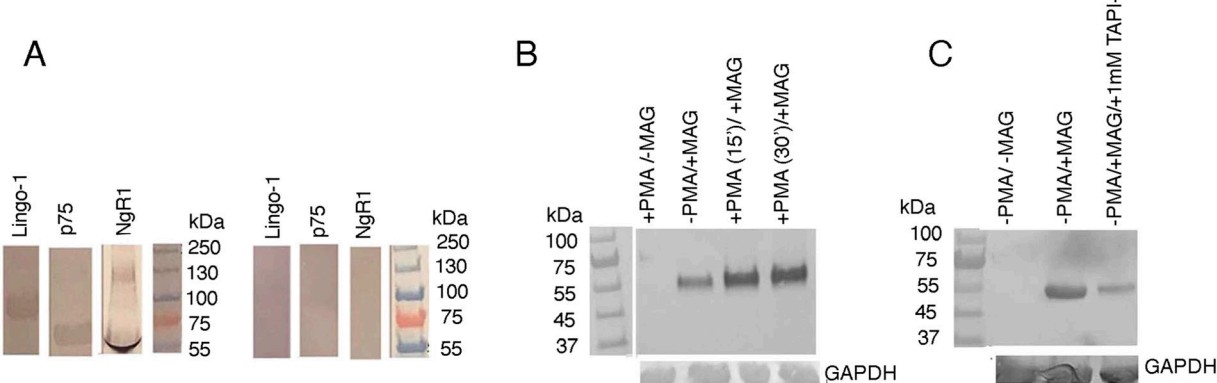

**Figure 5. ADAM17-mediated p75 ectodomain shedding in NG108-15 cells.**
**(A)** Endogenous expression of Lingo-1, p75, and NgR1 in the NG108-15 cell line before and after differentiation. The NG108-15 cells were differentiated with 1 mM dibutyl cAMP, which resulted in enhanced expressions of NgR1, Lingo-1, and p75. The cells were harvested, lysed, and subjected to Western blots analysis. For NogoR1, we detected minor higher molecular weight bands. This could be because of the association of membranous glycolipids, such as gangliosides, with NogoR1 in cell lysates of NG105-15, that slow the migration of NogoR1 on SDS–PAGE, or may be because of differently glycosylated species. The main 60 kD band represents the NgR1 receptor ectodomain. **(B)** PMA induced shedding of p75 in the presence or absence of the myelin inhibitor MAG. Briefly, $1 \times 10^5$ NG108-15 cells were differentiated by dibutyl cAMP (1 mM) for 3–4 d in six-well cell-culture plates (Nunc). The cells were transferred to conditioned media without FBS and allowed to grow for 24 h. PMA (25 ng/ml) was added to the media and incubated for either 15′ or 30′. Finally, the myelin inhibitor MAG (20 μg/ml) was added followed by an additional 30′ incubation. The culture supernatants (media and secreted proteins) were then harvested, concentrated, and subjected to Western Blot analysis using the p75 specific mAb 8J2. Likewise, the cells were lysed and blotted for the loading control, GAPDH. The results show that PMA augments the shedding of p75 in the presence of the exogenously added myelin inhibitor, MAG. **(C)** The ADAM17 inhibitor TAPI-1 causes significant inhibition of p75 shedding in absence of PMA. Differentiated NG108-15 cells, in absence of PMA, were incubated for 30′ with 1 mM TAPI-1 (30) before the addition of MAG. MAG was then added, followed by another 30′ of incubation. The supernatants were harvested and processed as before.

any regeneration efficacy in animal models (36). A humanized version of an anti-LINGO-1 antibody (37) was generated by Biogen Idec Inc., with clinical trials yielding reportedly promising preliminary results, but the outcome was not disclosed. Our studies provide a new approach that of targeting the p75 alpha secretase ADAM17 with inhibitory antibodies, which could now be evaluated in animal models of spinal cord injury, such as ex vivo organotypic slice cultures that essentially monitor biological response to stab, transection, or contusion injuries (38). In the long term, this could lead to the development of novel, alpha-secretase–targeting, therapeutic antibodies that promote neuronal regeneration in the spinal cord, providing a new approach for the treatment of spinal cord injury and paralysis.

# Materials and Methods

## Bacterial, mammalian, insect-cell, and neuroblastoma strains cell-culture conditions

*Escherichia coli* DH5α cells, grown in LB medium in the presence of 100 μg/ml ampicillin, were used for cloning. The mammalian HEK293 cell line, grown in DMEM-HG in the presence of 1% penicillin/streptomycin and 10% FBS, was used for expression of protein constructs. The stably transfected lines were maintained in the presence of 200 μg/ml hygromycin. The HEK293 cells were cultured at 37°C in the presence of 5% $CO_2$. The insect cell line Hi-5, grown in SF900 medium containing 1% penicillin/streptomycin

**Figure 6. Effect of ADAM and MMP inhibitors on p75 ectodomain shedding in NG108-15 cells.**
**(A)** Inhibition by the D8P1C1 and D5P2A11 mAbs. **(B)** Inhibition by the C12 mAb. Small-molecule inhibitors, including Batimastat (MMP inhibitor), TAPI-1 (ADAM17 and MMP inhibitor), and GI254023X (ADAM10 inhibitor), as well as 1H5 (inhibitory anti-ADAM10 mAb), were included in the assays for comparison. The NG108-15 cells were differentiated and incubated with mAbs or small-molecule inhibitors for 30'. This was followed by the addition of 25 ng/ml of PMA (30' incubation). Finally, MAG (20 μg/ml) was added, incubated for 30', and the supernatants were processed as before. The results show that the anti-ADAM17 mAbs, which target either the protease or the D+C domain region of ADAM17 (Fig 1A), abrogate p75 shedding at 20 μg/ml. **(C)** The percent inhibition of p75 shedding by the anti-ADAM17 mAbs was quantitated using the software Image J and GAPDH as the loading control. The bar graph demonstrates mean percent inhibition for each concentration of the inhibitors and the whiskers indicate ± SEM, n = 2. **(D)** Table showing the inhibitory potencies of the small-molecule inhibitors and the anti-ADAM mAbs. The percent inhibition of p75 shedding by the alpha secretase inhibitors (anti-ADAM10 or ADAM17 mAbs or small-molecule inhibitors) was quantitated using Image J software as follows: [1-{cleaved p75 in presence of myelin inhibitor MAG + inhibitors, normalized with respect to the loading control GAPDH}/{cleaved p75 in presence of MAG only, normalized with respect to GAPDH}]X100 (the amount of cleaved protein is estimated by the optical intensity of the corresponding SDS–PAGE band). GAPDH was used as a loading control (5). This table shows the average percent inhibition ± SEM for all inhibitors and controls (concentration as indicated in parentheses). Percent inhibition for all inhibitors and controls was analyzed using descriptive statistics. Statistical analyses were performed using IBM SPSS version 30 and GraphPad Prism version 10.

and cultured at 27°C, was also used for protein production. NG108-15 is a hybrid cell line formed by fusing rat glioma with mouse neuroblastoma cells, which has been used extensively to study p75 shedding and neurite outgrowth (2, 3). NG108-15 (ATCC) cells were cultured in DMEM without sodium pyruvate. The following components were added to the medium: 0.1 mM hypoxanthine 400 nM aminopterin, 0.016 mM thymidine, 10% FBS, 1.5 g/l sodium bicarbonate, and 1% penicillin and streptomycin.

**Expression and purification of the protein constructs**

Ganglioside GT1b trisodium salt (CAS 59247-13-1) was purchased from Santa Cruz Biotechnology and used for the entire study. The Na salt version of GT1b is water-soluble. The cDNA for NgR1 was a gift from Dr. Strittmatter (Yale) (20). The cDNAs for MAG (human), p75 (human) and LINGO-1 (mouse) were procured from Gene-script. The constructs human NgR1 (30–430), human p75 (31–210)

or (31–250), human MAG (22–504) and mouse Lingo-1 (35–516) were cloned in custom-made pcDNA 3.1⁺ vector (3) with a removable Fc tag at the C-terminus. The protein constructs were expressed in HEK293 cells and purified through protein A Sepharose, followed by size-exclusion chromatography on SD-200 (3). The human ADAM17 extracellular domain (ECD, 20–650 with the active-site E406 mutated to A) and ADAM17 D+C domains (474–646) were cloned in a custom-made baculovirus vector pMA152a fused to a removable Fc tag at the C-terminus (18). These two constructs were expressed and purified from baculovirus-infected Hi-5 cells and purified using protein A Sepharose, followed by size-exclusion on SD-200. For the ADAM17 ECD, the prodomain was cleaved off, and the mature polypeptide consisted of MP, D and C domains (215–650). The bovine ADAM10 constructs were expressed and purified from HEK293 cells as described before (12). All DNA constructs were sequenced to check for unwarranted PCR-generated mutations and the purified proteins were verified by N-terminal sequencing.

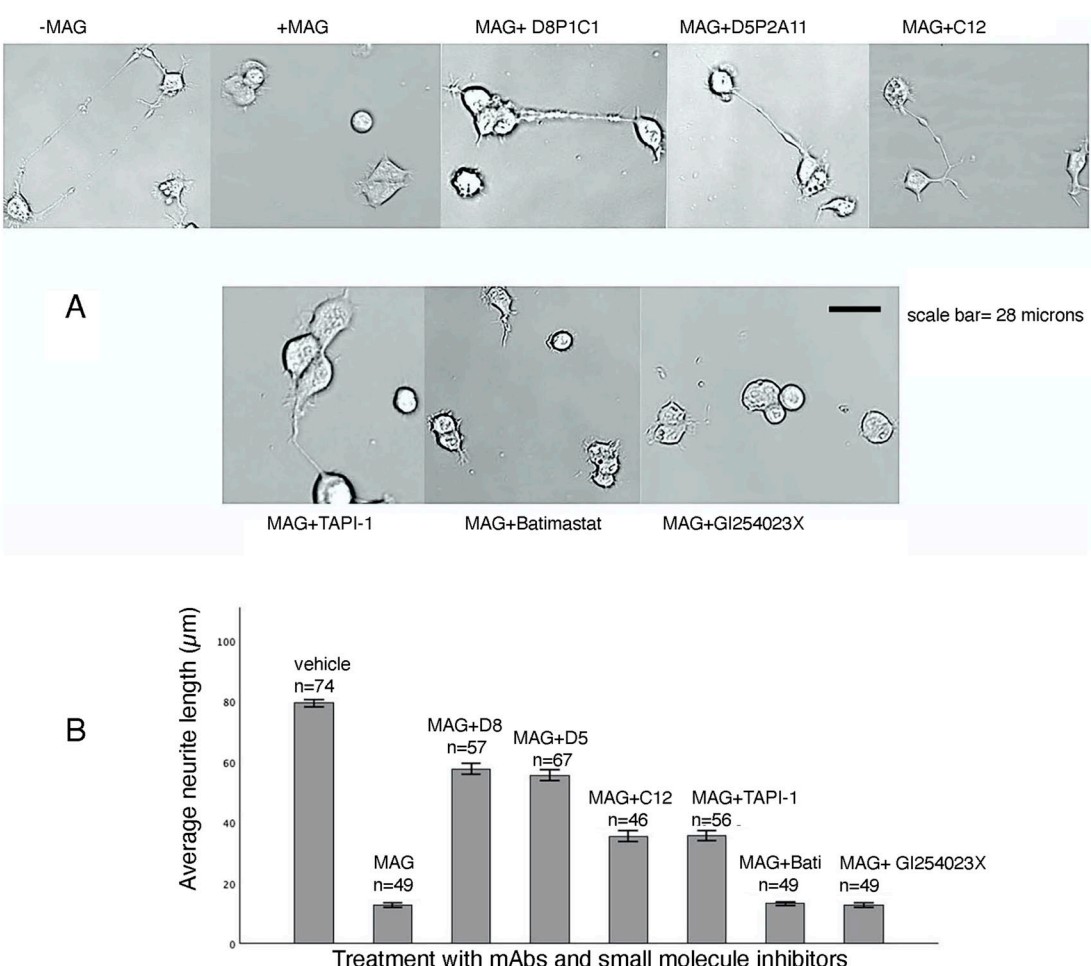

**Figure 7.   Effect of ADAM and MMP inhibitors on neurite outgrowth assays.**
**(A, B)** Differentiated NG108-15 cells were treated with inhibitors as indicated (A) and the longest primary neurite from each set was measured for 45–75 neurons (B). The average neurite length in µm is plotted (y-axis). Vehicle represents the well with no MAG, D8 represents D8P1C1, D5 represents D5P2A11, and Bati represents Batimastat. For more details, see Tables S1 and S2.

## Binding assays

### *In vitro pull-down experiments*

Biochemical pull-down assays were carried out using protein A-Sepharose beads (3). NgR1 and p75 were pre-incubated with GT1b-Na at a molar ratio of 1:10. In the pull-down experiments, Lingo-1-Fc was used as a "bait," whereas the untagged proteins (NgR1, p75, MAG) were used as "prey." The Lingo-1-Fc (bait) was incubated with protein A-Sepharose in 10 mM Hepes (pH 7.4), 150 mM NaCl, 0.05% Triton-X100 (binding buffer). The "prey" or candidate proteins were added and incubated at 25°C for 1 h. The protein A-Sepharose bound fractions were washed twice with the binding buffer and analyzed by SDS–PAGE (12%) under reducing conditions. In the pull-down assays, we used the shorter version of p75, residues 31–210, which includes all four cysteine-rich domains, because the longer version (21–250, see below) is not easily distinguishable from NgR1 on SDS–PAGE. The binding of ADAM17 ECD and D+C to the 5-mer complex was monitored in the similar fashion. Binding of ADAM17 D+C to isolated p75 was determined by immobilizing the p75-Fc–tagged version on protein A-Sepharose beads whereas binding of ADAM17 ECD to p75 was carried out using untagged p75 and ADAM17 ECD

tagged to Fc at the C-terminus. This is because ADAM17 ECD and p75-Fc show identical mobility on SDS–PAGE (reducing conditions).

### *Size-exclusion chromatography to study the interactions of NgR1, Lingo-1, p75 in the presence and absence of GT1b-Na salt*

The purified ectodomains, Lingo-1 (35–516), NgR1 (30–430), and p75 (31–250), were mixed at a molar ratio of 1:1:1 in the presence and absence of GT1b-Na. Size-exclusion chromatography was performed on a SD-200 column previously calibrated with a set of protein standards. For the size-exclusion chromatography studies, we used the longer version of p75 (31–250) that includes the entire ectodomain. The elution profiles of the individual proteins in the presence of GT1b-Na were also determined. The peak fractions were analyzed on 12% SDS–PAGE under reducing conditions.

### *BLI to quantitate the binding of ADAM17 ECD (E406A, active-site mutant) or ADAM17 D+C to the 5-mer complex or to isolated p75*

The 5-mer complex was immobilized on protein A sensors via Lingo-Fc. For the binding experiments to isolated p75, we used a p75-Fc–tagged construct. Specifically, protein A biosensors were loaded

with 50 µg/ml solution of purified 5-mer complex or p75-Fc, washed with HBS (20 mM Hepes, 150 mM NaCl pH 7.4), followed by addition of the ADAM proteases. Affinities (Kd) were calculated using the BLItzPro software (39).

### ELISA

The binding of the 5-mer complex or the isolated p75 to immobilized ADAM17 or ADAM10, ECD or D+C constructs, was evaluated by ELISA. Briefly, wells were coated with 100 µl (2 µg/ml) of ADAM17 ECD (E406A), ADAM17 D+C, ADAM10 ECD (E384, active-site mutant), or ADAM10 D+C overnight at 4°C. After three washes with PBS, pH 7.4, the wells were blocked with 4% non-fat dry milk. 100 µl of varying concentrations of the purified 5-mer complex (described in the Results section), or of isolated p75, were added to the wells and incubated for 1 h at RT. The p75 bound to the ADAM17 or ADAM10 constructs was detected by mouse mAb to the human p75 NGF receptor ectodomain (8J2, 1:100 dilution; Abcam). To detect the 5-mer complex bound to ADAM17 ECD/D+C, we used a mouse mAb specific to the human MAG ectodomain (MAG A-11, 1:100 dilution; Santa Cruz). Rabbit anti-mouse cross-adsorbed antibody conjugated to HRP (1:2,000 dilution; Invitrogen) was used as a secondary. Color was developed using the TMB substrate kit (Thermo Fisher Scientific), and data were recorded at 450 nm. Likewise, we used ELISA-based assay to gauge the effect of the three anti-ADAM17 mAbs D8P1C1, D5P2A11 (18), and C12 (28) on the binding of the 5-mer complex to immobilized ADAM17 ECD (E406A) or ADAM17 D+C. Varying concentrations of the three mAbs (D8P1C1, D5P2A11, or C12) were added to the wells and incubated for 1 h at RT. The wells were washed thrice and incubated with 100 µl of the purified 5-mer complex (25 µg/ml). The 5-mer complex bound to ADAM17 ECD/D+C was detected by MAG A-11.

### Endogenous expression of Lingo-1, p75, and NogoR1 in the NG108-15 cell line before and after differentiation

NG108-15 is a hybrid cell line formed by fusing the rat glioma with mouse neuroblastoma cells. This line has been used extensively to study p75 shedding and neurite outgrowth (2, 5). NG108-15 (ATCC) cells were cultured in DMEM without sodium pyruvate. The following components were added to the medium: 0.1 mM hypoxanthine, 400 nM aminopterin, 0.016 mM thymidine, 10% FBS, 1.5 g/l sodium bicarbonate, and 1% penicillin and streptomycin. The cells were differentiated with 1 mM dibutyl cAMP. Differentiated NG108-15 cells, a model system for studying cholinergic neurons, exhibited enhanced expression of NgR1, Lingo-1, and p75 (29, 31). The cells were harvested, lysed, and subjected to Western blots analysis. To detect the NgR1, we used the rabbit mAb, ARC0872 (1:100 dilution; Thermo Fisher Scientific) to the human NgR1 ectodomain that recognizes both human and mouse NgR1. Lingo-1 was detected by a rabbit polyclonal mAb to mouse Lingo-1 (extracellular antibody ANT-032, 1:100 dilution; Alomone labs). This polyclonal antibody can recognize human, mouse, and rat Lingo-1. We used the goat anti-rabbit IgG AP conjugate (1:1,000 dilution; Promega) as a secondary antibody for both NgR1 and Lingo-1. To detect p75 expression, we used the mouse 8J2 mAb described above (1:100 dilution) as a primary and an anti-mouse IgG AP conjugate (1:1,000 dilution; Promega) as a secondary. For GAPDH, we used the GAPDH loading control mAb (GA1R, 1:100 dilution) as a primary and an anti-mouse IgG AP conjugate as a secondary (1:1,000). Color was developed using NBT/BCiP as a substrate.

### The shedding of p75 from NG108-15 cells in the presence (or absence) of the myelin inhibitor MAG, ADAM protease inhibitors, and PMA

Briefly, $1 \times 10^5$ NG108-15 cells were differentiated by dibutyl cAMP (1 mM) (31) from 3–4 d in six-well cell-culture plates (Nunc). The cells were transferred to conditioned media without FBS and allowed to grow for 24 h. PMA (25 ng/ml) (25, 40) was added to the media and incubated for either 15' or 30'. Finally, the myelin inhibitor MAG (20 µg/ml) was added, followed by further incubation of 30'. The culture supernatants were harvested, concentrated, and subjected to Western Blot analysis using the p75 specific mAb 8J2. Likewise, the cells were lysed and blotted for the loading control, GAPDH. Control wells with +PMA/−MAG or −PMA/+MAG (constitutive cleavage) were included in the assay. For the wells indicated by +PMA/−MAG or −PMA/+MAG, PMA or MAG was added and incubated for 30' before the supernatants were harvested. Differentiated NG108-15 cells, in the absence of PMA, were treated with 1 mM TAPI-1 before the addition of MAG and incubated for 30'. MAG was added, followed by another 30' of incubation. The supernatants were harvested and processed as before.

### Inhibition of p75 ectodomain shedding by the anti-ADAM specific inhibitors and the MMP inhibitor batimastat (33)

The NG108-15 cells were differentiated and incubated with mAbs or small-molecule inhibitors for 30'. This was followed by the addition of PMA (30' incubation). Finally, MAG was added, incubated for 30', and the supernatants were processed as before. The table (Fig 6D) shows the inhibitory potencies of the small-molecule inhibitors and the anti-ADAM mAbs.

### Neurite outgrowth assays

$5 \times 10^4$ differentiated NG108-15 cells were plated in 8-chamber tissue culture slides (LabTek chambered #1.5 german coverglass system) (5). The cells were grown overnight for 16 h. PMA (25 ng/ml) was added to the wells. MAG (25 µg/ml) was mixed with different ADAM protease inhibitors in DMEM medium and added to the wells. We used 20 µg/ml of the ADAM mAbs and 1 µM of the small-molecule inhibitors. Cells were incubated for 4–6 h and the wells were imaged under an inverted Zeiss AxioObserver Z1 (Zeiss), 0.35 NA, 20x magnification. The images were acquired using Zen Blue 2.3. After systematic scanning of the 8-chamber slide, we acquired mutiple czi files for each condition (vehicle control, MAG, MAG+-inhibitors). From the mutiple image files assigned for each condition, we selected an average 45–75 individual neuronal cells in total and determined the lengths of the longest primary neurite using the Image J software with the Neuron J plugin (5) and a semi-manual script (written by Eric Chan, Molecular cytology core facility, MSKCC).

### Quantification and statistical analysis

The data from all in vitro and cell-based assays are representative of triplicate determinations. Statistical analysis was performed using IBM SPSS version 29.

**Key resource table**

| Cell lines/strains | Source | Identifier |
| --- | --- | --- |
| E coli DH5 | Thermo Fischer scientific | catalog number 18265017 |
| NG108-15 | ATCC | catalog number HB12317 |
| HEK293 | Millipore Sigma | catalog number 8512602-DNA-5UG |
| Hi-5 | Millipore Sigma | catalog number SRP3133-10UG |
| **Chemicals** | | |
| GT1b trisodium salt | Santa Cruz Biotech | catalog number CAS 59247-13-1 |
| SF900 11 SFM | Thermo Fischer scientific | catalog number 10902088 |
| PMA | Thermo Fischer scientific | catalog number AC356150010 |
| Dibutyl cAMP | Thermo Fischer scientific | catalog number CAS 16980-89-5 |
| DMEM-HG | MSKCC core | |
| Penicillin/streptomycin | MSKCC core | |
| RPMI | MSKCC core | |
| Fetal Bovine serum | Avantar | catalog number 97068-085 |
| Aminopterin | Thermo Fischer scientific | catalog number AC468751000 |
| Thymidine | Thermo Fischer scientific | catalog number AC226740050 |
| **Software and algorithms** | | |
| Image J software with Neuron J plugin | https://imagej.net/ij/, 1997-2018 | |
| IBM SPSS version 29 | | |

# Data Availability

For inquiries regarding materials/cell-lines/ protocols/plasmids and other reagents, please contact the lead contact: DB Nikolov at nikolovd@mskcc.org.

# Supplementary Information

# Acknowledgements

This work was supported by the New York State Spinal Cord Injury Research Program grant SCIRB funding, DOH-1 Part 5 IDEA award (2022-2024), and NIH grant R21AG080685 (2022-2024) to DB Nikolov, and NIH Cancer Center Support Grant (P30 CA008748) to Memorial Sloan Kettering Cancer Center.

## Author Contributions

N Saha: conceptualization, investigation, and writing—original draft, review, and editing.
E Chan: software and methodology.
RP Mendoza: software.
Y Romin: methodology.
MJ Tipping: supervision and methodology.
DB Nikolov: software, formal analysis, supervision, funding acquisition, methodology, and writing—original draft, review, and editing.

## Conflict of Interest Statement

The authors declare that they have no conflict of interest.

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
