## [Reviewer comments · Life Science Alliance]

Life Science Alliance

Antibodies targeting ADAM17 reverse neurite outgrowth inhibition by myelin- associated inhibitors.

Nayanendu Saha, Eric Chan, Rachelle Mendoza, Yevgeniy Romin, Murray Tipping, and Dimitar Nikolov

DOI: <https://doi.org/10.26508/lsa.202403126>

Corresponding author(s): Nayanendu Saha, Memorial Sloan Kettering Cancer Center and Dimitar Nikolov, Memorial Sloan Kettering Cancer Center

Review Timeline:

Submission Date:	2024-11-05
Editorial Decision:	2025-01-02
Revision Received:	2025-03-10
Editorial Decision:	2025-03-11
Revision Received:	2025-03-13
Accepted:	2025-03-14

Transaction Report:

January 2, 2025

Re: Life Science Alliance manuscript #LSA-2024-03126-T

Dr. Nayanendu Saha
Memorial Sloan-Kettering Cancer Center
Structural biology department
1275 York Avenue
New York, New York 10021

Dear Dr. Saha,

Thank you for submitting your manuscript entitled "Monoclonal antibodies targeting the alpha secretase ADAM17 block intramembrane proteolysis of p75 and reverse neurite outgrowth inhibition by myelin-associated inhibitors." to Life Science Alliance. The manuscript was assessed by expert reviewers, whose comments are appended to this letter. We invite you to submit a revised manuscript addressing the Reviewer comments.

Thank you for this interesting contribution to Life Science Alliance. We are looking forward to receiving your revised manuscript.

Sincerely,

B. MANUSCRIPT ORGANIZATION AND FORMATTING:

Reviewer #1 (Comments to the Authors (Required)):

The manuscript by Saha and co-authors, titled "Monoclonal antibodies targeting the alpha secretase ADAM17 block intramembrane proteolysis of p75 and reverse neurite outgrowth inhibition by myelin-associated inhibitors," presents compelling data on the role of blocking monoclonal antibodies (mAbs) that target the metalloproteinase ADAM17. These antibodies are shown to influence the regulated intramembrane proteolysis (RIP) of the low-affinity NGF receptor p75, a crucial process that impedes axon regeneration in response to myelin-derived inhibitors like MAG.

The study employs a reproducible biochemical in vitro assay that involves the formation of a 5-mer neuron-myelin signaling complex. This complex is initiated via GT1b-mediated preassembly of NgR1/Lingo-1, which subsequently binds to p75 when exposed to the myelin-derived inhibitor MAG. By utilizing this assay alongside pull-down experiments, ELISA-based binding assays, and Bio-layer interferometry, the authors identified a binding interaction between ADAM17 and p75 within the preassembled 5-mer neuron-myelin signaling complex, while no such interaction was observed with ADAM10. They also characterized the strength of this interaction and demonstrated that a specific group of mAbs targeting the C+D regions of ADAM17 effectively blocked this binding, with no effects noted from mAbs targeting ADAM10.

Furthermore, the authors established ADAM17 as the key alpha secretase responsible for the MAG-mediated RIP of p75 at the membrane of the differentiated NG108-15 neuroblastoma cell line. This was evidenced by the increased shedding of p75 into the extracellular environment and enhanced neurite outgrowth in response to mAbs specific to this metalloproteinase.

Overall, the manuscript highlights the potential therapeutic role of two anti-ADAM17 mAbs (D8P1C1 and D5P2A11), which target the protease's active site, in promoting axon regeneration across various human pathological conditions. While the manuscript provides significant insights into the characterization of blocking mAbs targeting ADAM17, several major concerns warrant attention.

Major Concerns

1. Throughout the manuscript, the western blot images lack quantification and statistical analysis.
2. In relation to Figure 2C, F, and G, the authors should further discuss the discrepancy observed in the binding of ADAM17ECD versus the D+C domains to p75. Specifically, why are the D+C domains in the ECD insufficient to maintain interaction with p75?
3. Data demonstrating the selective binding of the C12 mAb to the cysteine-rich domain of ADAM17 should be included.
4. Figure 3B is missing data regarding the effects of two mAbs on the binding of the neuron-myelin complex to immobilized ADAM17 D+C.
5. In Figure 3A, the authors state the data are derived from "triplicate determinations from two independent experiments." They should explain why a third independent experiment was omitted.
6. The immunodetection of GADH in the loading control bands from Figure 4B is unclear and should be repeated for clarity.
7. The NogoR lane in Figure 4A displays additional bands at unexpected molecular weights; this discrepancy needs clarification.
8. The paper would significantly benefit from incorporating primary neuronal cultures (e.g., granular cell cultures or dorsal root ganglion) to further validate the blocking efficacy of the anti-ADAM-17 mAbs in response to MAG.

Minor Concerns

1. In Figure 1B, the molecular weight standards overlap with bands from the eluate of Lingo-1.
2. In Figure 1C, arrows indicating the presence of NogoR are included, but they are not mentioned in the figure legend or the text. The figure would benefit from labeling p75 mobility in the eluate with arrows.
3. In several figures, labels indicating lane composition are either too close to the gel or partially overlapping, making them difficult to read.
4. Typographical error: "Here, the pre-formed the 5-mer complex" should be corrected.
5. Please replace "u" with "μ" where appropriate.
6. In Figure 6A, cells treated with MAG-TAP-1 appear to have a higher magnification compared to those treated with MAG+Batimastat; this should be reviewed.

Reviewer #2 (Comments to the Authors (Required)):

Reviewer's comments concerning „Monoclonal antibodies targeting the a-secretases ADAM10 block intramembrane proteolysis of p75 and reverse neurite outgrowth inhibition by myelin-associated inhibitors" by Saha and colleagues.

The present study addresses the role of regulated intramembrane proteolysis (RIP) for the regulation of a protein complex composed of p75, NgR1 and distinct co-receptor proteins that are involved in the detection of myelin associated axon outgrowth inhibitors. Using monoclonal antibodies the authors conclude that ADAM17 inactivates recognition of growth inhibitors via proteolysis of p75, thus permitting axon outgrowth in inhibitory environments. The work is highly relevant for the field of regenerative medicine.

The binding studies based on pull-down assays using protein A-Sepharose beads and Fc-constructs are convincing, as well as the cleavage and release studies of p75 55 kD fragment in NG108-15 cells.

1) The neurite outgrowth assay is less convincing. In the images shown the distinction of individual neurites is not obvious, as fibers from neighboring cells contact each other. Thus, it seems difficult to assess individual fibers. A neurite growth inhibition assay using primary neurons, for example from E18 rat hippocampus, would be more convincing.

2) A plot of the individual fiber lengths (% length distribution) would be desirable.

3) MAG occurs in membrane bound form in vivo, would substrate-adhered MAG also inhibit neurite outgrowth in the model? Here, the molecule is added as soluble compound to the culture medium.

4) Why have the neurites not been stained, e.g. with antibodies to β -tubulin to visualize faint fibers?

5) Details how the inhibition values were calculated in the molecular binding and the neurite outgrowth assays are missing.

Reviewer #3 (Comments to the Authors (Required)):

The manuscript by Saha and colleagues describes two sets of experiments. The first is the reporting that in vitro co-association of the MAG receptor of p75 NogoR1 and Lingo1 requires trisialoganglioside GT1b as part of the receptor complex for ligand(MAG)-induced alpha cleavage by ADAM17. The second reported finding is that particular antibodies directed at the p75 cleavage enzyme ADAM17 can suppress cleavage and functional activity of neurite inhibition. The aspect of where the antibodies are directed to and thus their abilities are unclear - block the enzyme, the enzyme doesn't work, and so? The two parts of the study are related, but they do not hang together well. This may be partly because of the combined results discussion and a lack of firm conclusions. I suggest this is rewritten.

There are also various uncertainties around methods, rationale, analysis and/or experimental numbers, so I find the story unconvincing and incomplete at this stage, but it has the potential to be of interest to a niche audience.

Specifics.

1. The finding of GT1b being part of the receptor complex is already reported over 10 years previously by the same group.
2. Fig 1 should have a diagram of the different portions of the receptors used in the study.
3. Fig 1 A should display size markers the lanes seem to be skewed in lanes 1-5/6 but not lanes 7/8 is this the same gel?
4. Fig 2B it is not clear what the volume refers to and how the numbers above the gel relate to the gel filtration image.
5. It is not clear what Fig 1C is demonstrating as fraction 13 would be expected to have all 3 proteins regardless of GT1b.
6. With GT1b, Nogo is mostly in fraction 14 as seen in 1B in the absence of co-receptors. Is Nogo in Fig 1C the same as NgR1?
7. The gels do not show p75 in lane 1 which is presumed to be the input to the gel filtration?
8. Fig 2A why is p75 at 30kDa? Is that only the ectodomain of p75? Fc-p75 also only seems to be the ectodomain.
9. It is also not clear what ADAM-ECD mutant is and why it's a mutant. What is D+C of Adam 17/10?
10. Presumably these are purified proteins not constructs (plasmids)? Fig2D-G uses protein Sepharose to bind a 4mer protein complex (via FcLingo?) and then ADAM proteins were added. Does Fc-Lingo alone bind ADAM - that would be the right control rather than beads alone.
11. Only D+C appears to interact with p75 whereas its ECD interacts with other components of the complex, and this is independent of MAG. This suggests cleavage could occur independently of ligand/Mag. Is this a problem for the hypothesis/model?
12. There is no conclusion or discussion on this aspect of the study.
13. Part 2
14. It would be good to understand the structure of ADAM17 in a diagram and where the different Antibody epitopes are.
15. C12 partially blocks binding of either ADAM fragment to p75/receptor complex- this seems to be a non-specific effect - what is the control e.g. another antibody to NogoR or p75?
16. Are any of the 5mer proteins Fcs in experiments resulting in Fig 3?
17. In Fig 3B is only C12 used or all 3 antibodies together?
18. Figure legend is unclear, N is not reported (For fig 3 and for most figures).
19. The protease domain appear to not be involved in binding the complex: how could this be right if the enzymatic region needs to contact p75 stalk to cleave it? Perhaps it binds then cuts only when MAG is there - is this supported by the evidence? Where and to what is it binding? If you take out each protein of the complex or test binding to each 2/3mer combination what do you see? Does p75 have to be membrane-bound/have a TM to interact with ADAM17?

20. Given the majority of the coreceptors are embedded in to attached to the membrane how is the interaction of purified proteins a good/justifiable replication of in vivo?
21. Fig 4 The GAPDH loading controls are of poor quality and should be from the same gels (also for Fig 5), and this is clearly not the case in Fig 4. Also different gels should not be directly adjacent and a line or gap should be obvious.
22. Why is the ECD of p75 being found inside cells rather than in the media? How much p75-ECD is in the media? One could probe for ICD p75 fragments on Westerns or use an ELISA for qualitative assessment for ECD p75 from the media (Biosensis). Perhaps TAPI only inhibits PMA-induced (or constitutive) cleavage and not MAG-induced cleavage, thereby accounting for the only partial inhibition? As indicated, it might it inhibit cleavage at the cell surface not in endosomes? And thus the assay is a poor measure of ADAM17 surface activity.
23. Fig 4C should not mention PMA as it is not added to any lane and it is confusing.
24. It is unclear why PMA is used at all, given a likely difference in mechanism of promoting cleavage - is there any evidence that PMA affects MAG expression or binding to p75 - rather than via an alternative route of cleavage enhancement (or via ADAM10)? It is not(?) used in subsequent experiments)
25. Fig 5 The use of different names for the antibodies (Ab1, Ab2) makes it difficult to understand the figures. The number of N independent experiments and replicates is unclear and the error bars are surprisingly tight (is it SEM or SD, this and N should be stated for all data presented). What is the level of cleavage without MAG (or PMA)?
26. Result text talks about "MAG-induced release of the ECD" but isn't PMA present?
27. Personal communication should not be used if it is the authors (that is 'data not shown' and why not?) otherwise the person from whom the communication is from and when should be identified.
28. The rationale of the experiments is unclear - why were the 3 antibodies and the inhibitors used? What was the question? The question could be whether the interaction between p75 and ADAM 17 - which might be inhibited by C12 - prevents cleavage of p75. In that case the other antibodies are controls for blocking ADAM17 activity (but not binding). But in that case, what are the other reagents being used to discover given they do not inhibit what they are "meant" to? Another way of inhibiting the interaction (e.g. lack of GT1b) might be used as a control.
29. The conclusion of part 2 is oblique, and not tied together with the results of Part 1.
30. The results are also inconsistent with Fig 6 results where C12 and TAPI have equivalent effects.
31. A non-related antibody should be used as a control.
32. The experimental analysis is problematic: In most of the pictures shown, the longest neurite length that could be measured is between 2 cells, and there are multiple smaller neurites. Was only the longest neurite measured? What was the definition of a neurite (2x the diameter of the cells is a typical definition); I am puzzled by the error bars (is this the average length from each of X independent experiments? or as it seems, the average of the written number of cells in one experiment).
33. What are the methods for Sup Fig 2?

We thank the reviewers for their valuable suggestions and comments. We have now revised the manuscript to fully address these. Below are listed all comments and the revisions that we made in response.

Reviewer #1

Major Concerns

1. Throughout the manuscript, the western blot images lack quantification and statistical analysis.

Response: We have now performed quantification and statistical analysis for the western blots on Figure 6 (old Figure 5). Figure 5 (old Figure 4) simply confirms previously published (reference 25) qualitative observations, so no quantitation was performed.

2. In relation to Figure 2C, F, and G, the authors should further discuss the discrepancy observed in the binding of ADAM17ECD versus the D+C domains to p75. Specifically, why are the D+C domains in the ECD insufficient to maintain interaction with p75?

Response: The discrepancy has been addressed on page 7: indeed, it was previously postulated that ADAM proteinases sample two distinct conformations: open, activated, where the D+C region is exposed for interactions with substrates (e.g. p75) and regulatory proteins, and closed, autoinhibited (the predominant conformation of the isolated ADAM17 ECD), where the D+C region interacts with and inhibits the ADAM catalytic (MP) domain¹⁶ (and is, thus, not available for interactions with p75). In addition, recent structural studies with the mature form of the ADAM17 ectodomain bound to iRhom2, showed a high degree of conformational flexibility in ADAM17²⁷. We, therefore, hypothesize that the ADAM17 interactions with the neuron-myelin signaling complex abet the mature alpha secretase to attain and maintain an activated/open conformation (*where the D+C region is exposed and available to bind p75*) leading to the RIP of p75. Such an activation mechanism would preclude the cleavage of p75 molecules that are not associated with signaling complexes, thus preventing aberrant downstream signaling.

3. Data demonstrating the selective binding of the C12 mAb to the cysteine-rich domain of ADAM17 should be included.

Response: The cryo-EM structure of C12 bound to the ADAM17 cysteine rich domain has now been published by our group and is available online. The reference (ref. 28) is now included in the bibliography section.

4. Figure 3B is missing data regarding the effects of two mAbs on the binding of the neuron-myelin complex to immobilized ADAM17 D+C.

Response: The two mAbs, D8P1C1 and D5P2A11, bind to the ADAM17 protease domain and do not interact with the D+C region of ADAM17 (reference 18). Since they specifically recognize a distinct epitope on ADAM17, completely absent in this assay, it is not necessary to evaluate how these two mAbs affect the interactions of the 5-mer complex with immobilized ADAM17 D+C.

5. In Figure 3A, the authors state the data are derived from "triplicate determinations from two independent experiments." They should explain why a third independent experiment was omitted.

Response: Each reading on old Fig. 4 (old Fig. 3) is a mean of three observations (n=3) (three independent wells were generated and measured). The experiment was repeated twice (each time in triplicate), and the two repetitions yielded identical results.

6. The immunodetection of GADH in the loading control bands from Figure 4B is unclear and should be repeated for clarity.

Response: GAPDH has been repeated (see New Fig. 5B).

7. The NogoR lane in Figure 4A displays additional bands at unexpected molecular weights; this discrepancy needs clarification.

Response: The concern has been addressed in legend to Figure 4A. "For Nogo-R1 we detected minor higher molecular weight bands. This could be due to the association of membranous glycolipids, such as gangliosides, with NogoR1 in cell lysates of NG105-15, that slow the migration of NogoR1 on SDS-PAGE or may be due to differently glycosylated species."

8. The paper would significantly benefit from incorporating primary neuronal cultures (e.g., granular cell cultures or dorsal root ganglion) to further validate the blocking efficacy of the anti-ADAM-17 mAbs in response to MAG.

Response: Our conclusion and interpretation of the work presented in the manuscript are coherent with the experiments performed and the cell lines used. Further validation of the activity of the anti-ADAM17 mAbs in primary neurons or mice models will be performed in future studies.

Minor Concerns

1. In Figure 1B, the molecular weight standards overlap with bands from the eluate of Lingo-1.

Response: Figure 1B has been redone.

2. In Figure 1C, arrows indicating the presence of NogoR are included, but they are not mentioned in the figure legend or the text. The figure would benefit from labeling p75 mobility in the eluate with arrows.

Response: The p75 mobility in Figure 1C has been labeled and the arrows have been described in the legend.

3. In several figures, labels indicating lane composition are either too close to the gel or partially overlapping, making them difficult to read.

Response: All figures have been redone to address this.

4. Typographical error: "Here, the pre-formed the 5-mer complex" should be corrected.

Response: The typographical error has been corrected.

5. Please replace "u" with " μ " where appropriate.

Response: The symbol has been replaced.

6. In Figure 6A, cells treated with MAG-TAP-1 appear to have a higher magnification compared to those treated with MAG+Batimastat; this should be reviewed.

Response: All the wells were imaged under an inverted Zeiss AxioObserver Z1 (Zeiss, Germany), 0.35 NA, 20x magnification.

Reviewer #2

1. The neurite outgrowth assay is less convincing. In the images shown the distinction of individual neurites is not obvious, as fibers from neighboring cells contact each other. Thus, it seems difficult to assess individual fibers. A neurite growth inhibition assay using primary neurons, for example from E18 rat hippocampus, would be more convincing.

Response: These studies were performed according to published protocols used in previous studies (reference 34, 35). We have now also included the lengths of the longest neurites from each set along with statistical significance (Supplementary Tables S2A and S2B). Our conclusion and interpretation of the work presented in the manuscript are coherent with the experiments performed and the cell lines used. Further validation of the activity of the anti-ADAM17 mAbs in primary neurons or mice models will be performed in future studies.

2) A plot of the individual fiber lengths (% length distribution) would be desirable.

Response: The lengths of individual fibers along with statistics have now been included as Supplementary Tables S2A and S2B.

3) MAG occurs in membrane bound form in vivo, would substrate-adhered MAG also inhibit neurite outgrowth in the model? Here, the molecule is added as soluble compound to the culture medium.

Response: The differences between membrane-bound v.s. soluble MAG need to be examined in co-culture models. In this manuscript, we performed the cell-based assays in accordance with previously established (by other groups) protocols (references 34 and 35) and used the purified ectodomain of MAG for activation of signaling.

4) Why have the neurites not been stained, e.g. with antibodies to β -tubulin to visualize faint fibers?

Response: We imaged the wells under an inverted Zeiss AxioObserver Z1 (Zeiss, Germany). We did not encounter any difficulties in detecting and measuring the clearly visible longest primary neurites from each set, so we did not need to use staining.

5) Details how the inhibition values were calculated in the molecular binding and the neurite outgrowth assays are missing.

Response: The details as to how the inhibition was calculated are now included in the legend to Figure 6D (old Figure 5D).

Reviewer # 3

Part1

1. The finding of GT1b being part of the receptor complex is already reported over 10 years previously by the same group.

Response: Our previous publication reported the role of GT1b during neuronal signaling initiated by a different myelin-associated inhibitor, Nogo. Here, we substantiate our previous findings for neuronal signaling initiated by another myelin-associated inhibitor MAG. In addition, we show for the first time that ADAM17, and not ADAM10, is the alpha secretase that recognizes and cleaves p75 when it is a part of a 5-component neuron-myelin signaling complex comprising of NgR1, Lingo-1, p75, GT1b and a myelin-associated inhibitor. Importantly, we demonstrate the ability of inhibitory anti-ADAM17 monoclonal antibodies to abrogate the cleavage of p75 in a neuroblastoma-glioma cell line and to reverse the neurite outgrowth inhibition by myelin-associated inhibitors.

2. Fig 1 should have a diagram of the different portions of the receptors used in the study.

Response: We have now included such diagram – New Fig 1A.

3. Fig 1 A should display size markers the lanes seem to be skewed in lanes 1-5/6 but not lanes 7/8 is this the same gel?

Response: Yes, it is the same gel and size markers' positions are now displayed on all figures.

4. Fig 2B it is not clear what the volume refers to and how the numbers above the gel relate to the gel filtration image.

Response: Volumes refer to elution volumes and numbers refer to the different fractions from SD-200 separation. This is also now clearly stated in the legend to Fig. 1.

5. It is not clear what Fig 1C is demonstrating as fraction 13 would be expected to have all 3 proteins regardless of GT1b.

Response: Figure 1C shows how the individual proteins migrate on an SD200 column. Figure 1D shows that in the presence of GT1b the complex containing all three proteins migrates at 10.5 ml (fraction 11). In the absence of GT1b, on the other hand, the elution profile of the mixture is identical to the sum of the elution profiles of the individual proteins documenting that the three proteins do not form a stable complex. The proteins now are highlighted with arrowheads on Fig. 1D.

6. With GT1b, Nogo is mostly in fraction 14 as seen in 1B in the absence of co-receptors. Is Nogo in Fig 1C the same as NgR1?

Response: Yes, NgR1 in Fig. 1 panels C and D is the same (it was mistakenly labeled as Nogo instead of NgR1 before). Fraction 14, in Fig. 1D (old Fig. 1C, in presence of GT1b), shows some excess unbound NgR1 that is not part of the Lingo-1, NgR1, p75, GT1b complex.

7. The gels do not show p75 in lane 1 which is presumed to be the input to the gel filtration?

Response: The position of the p75 protein is now indicated by reverse arrow (new Fig. 1D).

8. Fig 2A why is p75 at 30 kDa? Is that only the ectodomain of p75? Fc-p75 also only seems to be the ectodomain.

Response: We generated and used two p75 protein constructs (residues 31-210, which is a slightly truncated version) and (residues 31-250, which is the full ectodomain). In all our assays, both constructs behave identically, documenting that the 31-210 region is necessary and sufficient for the interactions with NgR1, Lingo-1 and ADAM17. In the biochemical pull-down assays (e.g., Fig 1B, Fig. 2 and Fig. 3), we used the 31-210 construct because it runs at a lower molecular weight position, around 30 kDa, and is, therefore, easier to separate from the NgR1 protein on the SDS gels. This is now explained in the main text of the manuscript.

9. It is also not clear what ADAM-ECD mutant is and why it's a mutant. What is D+C of Adam 17/10?

Response: For the ADAM ECD constructs, we used the active-site mutants ADAM17(E406A) and ADAM10(E384A) to prevent any unwarranted proteolysis. This is now explicitly stated in the main text and Materials and Methods of the revised manuscript. ADAM10 and ADAM17 (D+C) are the protein constructs comprising of the ADAM10/17 D and C domain region. This is now defined on page 3.

10. Presumably these are purified proteins not constructs (plasmids)? Fig. 2D-G uses protein Sepharose to bind a 4mer protein complex (via FcLingo?) and then ADAM proteins were added. Does Fc-Lingo alone bind ADAM - that would be the right control rather than beads alone.

Response: Yes, these are purified proteins. "Protein constructs" can generally refer to both the DNA that generates the protein, as well as the resulting protein. Yes, the pull down with protein-A Sepharose is via Fc-Lingo. Fc-Lingo alone does not bind ADAM10 or ADAM17.

11. Only D+C appears to interact with p75 whereas its ECD interacts with other components of the

complex, and this is independent of MAG. This suggests cleavage could occur independently of ligand/Mag. Is this a problem for the hypothesis/model?

Response: Various previous structural studies have revealed that the substrate-recognition ADAM D+C region is unavailable for interactions with substrate proteins in the autoinhibited/housekeeping ADAM conformation (where it interacts with the ADAM MP region). ADAM activation, which can occur in different ways, results in ADAM adopting an open conformation where the D+C region can more stably associate with ADAM substrates (such as p75) and position them for cleavage. Our current model (as elaborated on page 7 of the revised manuscript) is that interactions with the 4- and 5-component complexes induce the transition of ADAM17 from an autoinhibited/closed to an activated/open conformation. While this conformational transition, which by itself is MAG-independent, allows the ADAM ECD to stably bind the p75 substrate (via the D+C region), MAG is still needed for effective p75 cleavage (as evident from Fig. 5). The exact mechanism of how MAG facilitates p75 cleavage when bound to the activated/open ADAM17 conformation is not clear and high-resolution structural data of the 5-component complex is needed to determine the exact catalytic mechanism (see page 14 of the revised manuscript).

12. There is no conclusion or discussion on this aspect of the study.

Response: We have now formatted the manuscript as one single part (Results and Discussion) instead of two. Nonetheless, we do surmise the findings shown on Figs. 1, 2, and 3 (please see page 7 “*The results outlined above document that ...*”).

13. Part 2

14. It would be good to understand the structure of ADAM17 in a diagram and where the different Antibody epitopes are.

Response: We have included such diagram – New Fig 1A.

15. C12 partially blocks binding of either ADAM fragment to p75/receptor complex - this seems to be a non-specific effect - what is the control e.g. another antibody to NogoR or p75?

Response: We recently reported the cryo-EM structure of C12 bound to its target, the ADAM17 D+C domain region (reference 28). This blocking is a specific effect – C12 specifically binds to and prevents the ADAM D+C region (present in both ADAM constructs) from interacting with the p75/receptor complex. The controls are the D8 and D5 mAbs, which bind to the ADAM17 proteinase domain and do not block the binding of ADAM17 to the p75/receptor complex.

16. Are any of the 5mer proteins Fcs in experiments resulting in Fig 3?

Response: Lingo-1 is Fc tagged.

17. In Fig 3B is only C12 used or all 3 antibodies together?

Response: Only C12 is used in Fig. 4B (old Fig. 3B). ADAM17 D+C is used in Fig. 4B, which lacks the binding epitopes for the other two mAbs.

18. Figure legend is unclear; N is not reported (For fig 3 and for most figures).

Response: The legend is now updated, and N is included in the legends to new Figs. 2 and 4 (old Figs. 2 and 3).

19. The protease domain appear to not be involved in binding the complex: how could this be right if the enzymatic region needs to contact p75 stalk to cleave it? Perhaps it binds then cuts only when MAG is there - is this supported by the evidence? Where and to what is it binding? If you take out each protein of the complex or test binding to each 2/3mer combination what do you see? Does p75 have to be membrane-bound/have a TM to interact with ADAM17?

Response: The interaction between the ADAM17 proteinase domain and its target substrate sequence (p75 stalk) is transient, with a high off rate, and thus do not result in stable complex that can be resolved on a gel filtration column or in a pull-down assay. The current model is that the interactions between the ADAM17 D+C substrate-binding region and the p75 ECD are the high-affinity and high-specificity interactions that position the substrate sequence and the ADAM17 MP domain in proximity with each other and in orientation that can result in efficient cleavage. Our data indicates that MAG does not affect the affinity of the ADAM17(D+C)-p75(ECD) interactions but is essential for the catalytic reaction to proceed. P75 does not need to be membrane attached to interact with ADAM17. Elucidation of the exact mechanism of cleavage initiation in the presence of MAG needs high resolution structures of complexes along the catalytic reaction coordinate.

20. Given the majority of the coreceptors are embedded in to attached to the membrane how is the interaction of purified proteins a good/justifiable replication of in vivo?

Response: Our biochemical binding studies involve the protein ectodomains which interact with each other in solution, and this results in formation of biologically relevant complexes. There is no data to suggest that the transmembrane protein regions (or GPI anchors) are important for these interactions. Such observations are common for a lot of other cell surface receptor-co-receptor interactions and the literature is full of biochemical studies involving interacting ectodomains (and lacking the transmembrane regions). In addition, our studies include investigating the role of the head group of GT1b (which is one of the main constituents of the outer leaflet of the lipid bilayer) in the receptor-coreceptor ectodomain interactions.

21. Fig 4 The GAPDH loading controls are of poor quality and should be from the same gels (also for Fig 5), and this is clearly not the case in Fig 4. Also, different gels should not be directly adjacent and a line or gap should be obvious.

Response: A gap between the Western blots is now included in all figures of the revised manuscript. For Figures 5 and 6 (old Figures 4 and 5), the GAPDH loading control blots and the cleaved p75 ECD blots are different gels and they cannot possibly be in one gel: Indeed, the samples are centrifuged before performing the western blots and the supernatants (p75 released in the media) are resolved and botted (with anti-p75 mAb) on the upper gels, while the pellets (cell-associated GAPDH) are resolved and blotted (with anti-GAPDH mAb) on the lower gels.

22. Why is the ECD of p75 being found inside cells rather than in the media? How much p75-ECD is in the media? One could probe for ICD p75 fragments on Westerns or use an ELISA for qualitative assessment for ECD p75 from the media (Biosensis). Perhaps TAPI only inhibits PMA-induced (or constitutive) cleavage and not MAG-induced cleavage, thereby accounting for the only partial inhibition? As indicated, it might it inhibit cleavage at the cell surface not in endosomes? And thus the assay is a poor measure of ADAM17 surface activity.

Response: The ECD of p75 is not being found inside the cells. All throughout the manuscripts, the only p75 that is measured is the one cleaved off the cell surface and released into the cell media. All western blots are of the released in the media p75. TAPI does not only inhibit PMA-induced (or constitutive) cleavage, but also inhibits MAG-induced cleavage in the absence of PMA – see Figure 5C (this panel is done in the absence of PMA). The assay is an excellent measure of the ADAM17 surface activity because it directly measures the ADAM17-mediated release of p75 from the cell surface (and into the supernatants/media).

23. Fig 4C should not mention PMA as it is not added to any lane and it is confusing.

Response: Fig 5B and Fig 5C (old figures 4B and 4C) are meant to be looked at together (that is why they are next to each other) and the labeling underscores the fact that PMA is included in all lanes of panel C and in none of the lanes in panel D. This labeling also highlights the answer to the question of whether TAPI only inhibits PMA-induced cleavage (see question 22 above).

24. It is unclear why PMA is used at all, given a likely difference in mechanism of promoting cleavage - is there any evidence that PMA affects MAG expression or binding to p75 - rather than via an alternative route of cleavage enhancement (or via ADAM10)? It is not(?) used in subsequent experiments).

Response: We used PMA since it is known to enhance the cleavage of substrates by ADAM17 and is used in most similar studies. See for example references 34 and 35, which study the same MAG induced inhibitory signaling in similar cell lines, and where PMA is used. To the best of our knowledge, there is no evidence that PMA affects MAG expression or binding to p75 (or has any other alternative mode of action), as is also noted in references 34 and 35.

25. Fig 5 The use of different names for the antibodies (Ab1, Ab2) makes it difficult to understand the figures. The number of N independent experiments and replicates is unclear and the error bars are surprisingly tight (is it SEM or SD, this and N should be stated for all data presented). What is the level of cleavage without MAG (or PMA)?

Response: We have edited the figures and replaced "mAb1" and "mAb2" with the actual names of the mAbs. In Fig. 6 (old Fig. 5) we have included a table which lists the errors – these are SEM. N is now stated in the figure legend. N is also stated in all other figure legends for all data presented. There is no cleavage without MAG – see Figure 5B for the cleavage levels +/- PMA and +/- MAG.

26. Result text talks about "MAG-induced release of the ECD" but isn't PMA present?

Response: Without MAG there is no release of p75 ECD in the supernatant/media, even in the presence of PMA (Fig. 5). PMA (as used by us and many other groups: e.g., refs. 34, 35) only enhances the MAG-induced cleavage.

27. Personal communication should not be used if it is the authors (that is 'data not shown' and why not?) otherwise the person from whom the communication is from and when should be identified.

Response: We have now published the generation of the C12 mAb and the cryo-EM structure of C12 bound to the ADAM17 D+C region and included the new reference (reference 28).

28. The rationale of the experiments is unclear - why were the 3 antibodies and the inhibitors used? What was the question? The question could be whether the interaction between p75 and ADAM 17 - which might be inhibited by C12 - prevents cleavage of p75. In that case the other antibodies are controls for blocking ADAM17 activity (but not binding). But in that case, what are the other reagents being used to discover given they do not inhibit what they are "meant" to? Another way of inhibiting the interaction (e.g. lack of GT1b) might be used as a control.

Response: We used 3 antibodies, out of which two (D8P1C1 and D5P2A11) - bind to the ADAM17 protease domain that mediates the cleavage but not the high-affinity recognition and binding to the neuronal receptor/co-receptor complex, and one (C12) - binds to the substrate-recognition domain of ADAM17. The rationale was to investigate the potency of different ADAM17-targeting mAbs to inhibit the shedding of p75, either by interfering with the interaction of ADAM17 with the neuronal receptor/co-receptor complex or by blocking the ADAM17 protease domain. We included small molecule antagonists both for comparison of the mAbs' potential therapeutic potency, and in order to investigate the potential contributions of MMPs and of ADAM10 to p75 shedding and neurite outgrowth inhibition.

29. The conclusion of part 2 is oblique, and not tied together with the results of Part 1.

Response: We have now included a separate Conclusion section to address this (page 14). We have also edited the conclusion text, so it reads better and ties with the results of Part 1 (pages 14-15 of the revised manuscript).

30. The results are also inconsistent with Fig. 6 results where C12 and TAPI have equivalent effects.

Response: The C12 and TAPI effects are very similar in Fig. 6 (old Fig. 5) and Fig. 7 (old Fig. 6). In Fig. 7, C12 and TAPI have approximately the same effect, and in Fig. 6 the effect of TAPI is right in the middle between the effects of C12 at 10 ug/ml and 20 ug/ml (Fig. 6C).

31. A non-related antibody should be used as a control.

Response: As controls we used inhibitors of ADAM10 (GI254023X), ADAM17 (TAPI-1) and MMPs (Batimastat). These inhibitors are non-related to the D8/D5/C12 mAbs. We also used as a control the non-related antibody 1H5.

32. The experimental analysis is problematic: In most of the pictures shown, the longest neurite length that could be measured is between 2 cells, and there are multiple smaller neurites. Was only the longest neurite measured? What was the definition of a neurite (2x the diameter of the cells is a typical definition); I am puzzled by the error bars (is this the average length from each of X independent experiments? or as it seems, the average of the written number of cells in one experiment).

Response: Yes, only the longest neurite was measured. This experiment was performed exactly following the protocol (and neurite selection procedures/definitions) reported by others (see Domeniconi et. al., references 34 and 35). These well cited studies (references 34 and 35) are widely accepted by the scientific community. We have now included in the revised manuscript the complete raw data on neurite measurements and statistical analysis as Tables S2A and S2B (Supplementary Materials). Please also see the updated legend to Fig. 7A and Materials and Methods. For each cell, we measured the length of the longest primary neurite that it projects, and the error bars are for the average length measured for each experimental condition.

33. What are the methods for Sup Fig 2?

Response: Assuming the referee is referring to Supplementary Fig. S1, the method has been included in Materials and Methods, under Binding assays, section C: Bio-Layer Interferometry to quantitate the binding of ADAM17 ECD(E406A) or ADAM17 D+C to the 5-mer complex or to isolated p75.

March 11, 2025

RE: Life Science Alliance Manuscript #LSA-2024-03126-TR

Dr. Nayanendu Saha
Memorial Sloan Kettering Cancer Center
Structural biology department
430 east 67th street
NY, NY 10065

Dear Dr. Saha,

Thank you for submitting your revised manuscript entitled "Antibodies targeting ADAM17 reverse neurite outgrowth inhibition by myelin-associated inhibitors.". We would be happy to publish your paper in Life Science Alliance pending final revisions necessary to meet our formatting guidelines.

- please be sure that the authorship listing and order is correct
- please upload your main manuscript text as an editable doc file, without the line numbers
- please add ORCID ID for the secondary corresponding author- they should have received instructions on how to do so
- Please add a Category for your manuscript in our system
- please add the X and Bluesky handles of your host institute/organization as well as your own or/and one of the authors in our system
- Please remove figures from the manuscript file and leave them uploaded separately
- Please incorporate any points from the Conclusion section into the Discussion; we only allow a Discussion section
- please use the [10 author names, et al.] format in your references (i.e., limit the author names to the first 10)
- please add your main and supplementary figure legends to the main manuscript text after the references section
- we encourage you to revise the figure legend for figure S1 such that the figure panels are introduced in alphabetical order
- There is a legend for figure S2, but this figure has not been provided.
- Table S1 is missing
- Please upload your Tables in editable .doc or excel format. -Tables should be numbered consecutively with Arabic numerals (1, 2, 3, 4); They can be included at the bottom of the main manuscript file or be sent as separate files.
- please add callouts for Figures S1A-D to your main manuscript text

FIGURE CHECK:

-There are splices after the first columns in figure 5B and 6B. Please indicate the splice with a vertical black line and mention in the figure legends that the line indicates a splice in the blot.

A. FINAL FILES:

B. MANUSCRIPT ORGANIZATION AND FORMATTING:

Sincerely,

March 14, 2025

RE: Life Science Alliance Manuscript #LSA-2024-03126-TRR

Dr. Nayanendu Saha
Memorial Sloan Kettering Cancer Center
Structural biology department
430 east 67th street
NY, NY 10065

Dear Dr. Saha,

Thank you for submitting your Research Article entitled "Antibodies targeting ADAM17 reverse neurite outgrowth inhibition by myelin-associated inhibitors.". It is a pleasure to let you know that your manuscript is now accepted for publication in Life Science Alliance. Congratulations on this interesting work.

DISTRIBUTION OF MATERIALS:

Again, congratulations on a very nice paper. I hope you found the review process to be constructive and are pleased with how the manuscript was handled editorially. We look forward to future exciting submissions from your lab.

Sincerely,
